# The Set Autoencoder: Unsupervised Representation Learning for Sets

## Abstract

We propose the *set autoencoder*, a model for unsupervised representation learning for sets of elements. It is closely related to sequence-to-sequence models, which learn fixed-sized latent representations for sequences, and have been applied to a number of challenging supervised sequence tasks such as machine translation, as well as unsupervised representation learning for sequences. In contrast to sequences, sets are permutation invariant. The proposed set autoencoder considers this fact, both with respect to the input as well as the output of the model. On the input side, we adapt a recently-introduced recurrent neural architecture using a content-based attention mechanism. On the output side, we use a stable marriage algorithm to align predictions to labels in the learning phase. We train the model on synthetic data sets of point clouds and show that the learned representations change smoothly with translations in the inputs, preserve distances in the inputs, and that the set size is represented directly. We apply the model to supervised tasks on the point clouds using the fixed-size latent representation. For a number of difficult classification problems, the results are better than those of a model that does not consider the permutation invariance. Especially for small training sets, the set-aware model benefits from unsupervised pretraining.

## 1 Introduction

Autoencoders are a class of machine learning models that have been used for various purposes such as dimensionality reduction, representation learning, or unsupervised pretraining (see, e.g., Hinton & Salakhutdinov (2006); Bengio (2009); Erhan et al. (2010); Goodfellow et al. (2016)). In a nutshell, autoencoders are feed-forward neural networks which encode the given data in a latent, fixed-size representation, and subsequently try to reconstruct the input data in their output variables using a decoder function. This basic mechanism of encoding and decoding is applicable to a wide variety of input distributions. Recently, researchers have proposed a sequence autoencoder (Dai & Le, 2015), a model that is able to handle sequences of inputs by using a recurrent encoder and decoder. Furthermore, there has been growing interest to tackle sets of elements with similar recurrent architectures (Vinyals et al., 2015a; 2016; Xu et al., 2016).

In this paper, we propose the *set autoencoder* – a model that learns to embed a set of elements in a permutation-invariant, fixed-size representation using unlabeled training data only. The basic architecture of our model corresponds to that of current sequence-to-sequence models (Sutskever et al., 2014; Chan et al., 2016; Vinyals et al., 2015c): It consists of a recurrent encoder that takes a set of inputs and creates a fixed-length embedding, and a recurrent decoder that uses the fixed-length embedding and outputs another set. As encoder, we use an LSTM network with an attention mechanism as in (Vinyals et al., 2015a). This ensures that the embedding is permutation-invariant in the input. Since we want the loss of the model to be permutation-invariant in the decoder output, we re-order the output and align it to the input elements, using a stable matching algorithm that calculates a permutation matrix. This approach yields a loss which is differentiable with respect to the model's parameters. The proposed model can be trained in an unsupervised fashion, i.e., without having a labeled data set for a specific task.

In a series of experiments, we analyze the properties of the embedding. For example, we show that the learned embedding is to some extent distance-preserving, i.e., the distance between two sets of elements correlates with the distances of their embeddings. Also, the embedding is smooth, i.e., small changes in the input set lead to small changes of the respective embedding. Furthermore, we show

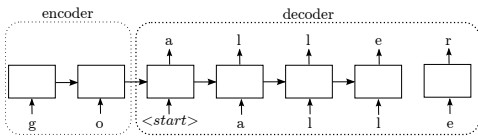

Figure 1: Example of a sequence-to-sequence translation model. The encoder receives the input characters ["g","o"]. Its internal state is passed to the decoder, which outputs the translation, i.e., the characters of the word "aller".

that pretraining in an unsupervised fashion can help to increase the performance on supervised tasks when using the fixed-size embedding as input to a classification or regression model, especially if training data is limited.

The rest of the paper is organized as follows. Section 2 introduces the preliminaries and briefly discusses related work. In Section 3, we present the details of the set autoencoder. Section 4 presents experimental setup and results. We discuss the results and conclude the paper in Section 5.

## 2 RELATED WORK

### 2.1 SEQUENCE-TO-SEQUENCE MODELS

Sequence-to-sequence models have been applied very successfully in various domains such as automatic translation (Sutskever et al., 2014), speech recognition (Chan et al., 2016), or image captioning (Vinyals et al., 2015c). In all these domains, the task is to model $P(Y|X)$, i.e., to predict an output sequence $Y = (y_1, y_2, \ldots, y_m)$ given an input sequence $X = (x_1, x_2, \ldots, x_n)$. Figure 1 shows the basic architecture of a sequence-to-sequence model. It consists of an *encoder* and a *decoder*, both of which are usually recurrent neural networks (RNNs). In the figure, the sequence-to-sequence model translates the input sequence $X = (g, o)$ to the output sequence $Y = (a, l, l, e, r)$. One by one, the elements of the input sequence are passed to the encoder as inputs. The encoder always updates its internal state given the previous state and the new input. Now, the last internal state of the encoder represents a fixed-size embedding of the input sequence (and is sometimes referred to as the *thought vector*). The decoder network's internal state is now initialized with the thought vector, and a special "start" token is passed as the input. One by one, the decoder will now output the tokens of the output sequence, each of which is used as input in the next decoder step. By calculating a loss on the output sequence, the complete sequence-to-sequence model can be trained using backpropagation. A special case of a sequence-to-sequence model is the sequence autoencoder (Dai & Le, 2015), where the task is to reconstruct the input in the output. For a more formal description of sequence-to-sequence models, please refer to (Sutskever et al., 2014).

### 2.2 SETS INSTEAD OF SEQUENCES IN INPUT OR OUTPUT

Researchers have tackled sets of elements directly with neural networks, without using explicit but lossy set representations such as the popular bag-of-words-model (Harris, 1954). Vinyals et al. raise the question of how the sequence-to-sequence architecture can be adapted to handle sets. They propose an encoder that achieves the required permutation-invariance to the input elements by using a content-based attention mechanism. Using a pointer network (Vinyals et al., 2015b) as decoder, the model can then be trained to sort input sets and outperforms a model without a permutation-invariant encoder. The proposed attention-based encoder has been used successfully in other tasks such as one-shot or few-shot learning (Vinyals et al., 2016; Xu et al., 2016). Another approach (Ravanbakhsh et al., 2016) tackles sets of fixed size by introducing a permutation-equivariant[1] layer in standard neural networks. For sets containing more than a few elements, the proposed layer helps to solve problems like point cloud classification, calculating sums of images depicting numbers, or set anomaly detection. The proposed models can fulfill complex supervised tasks and operate on sets of elements by exploiting permutation equi- or invariance. However, they do not make use of unlabeled data.

## 3 THE SET AUTOENCODER

The objective of the set autoencoder is very similar to that of the sequence autoencoder (Dai & Le, 2015): to create a fixed-size, permutation-invariant *embedding* for an input set $X =$

---

[1]A function $g$ is permutation equivariant, if $\pi(g(x)) = g(\pi(x))$, for all permutations $\pi$. However, we are more interested in permutation *invariant* functions $g$, such that $g(x) = g(\pi(x)), \forall \pi$

$\{x_1, x_2, \ldots, x_n\}, x_i \in \mathbb{R}^d$ by using unsupervised learning, i.e., unlabeled data. The motivation is that unlabeled data is much easier to come by, and can be used to pretrain representations, which facilitate subsequent supervised learning on a specific task (Erhan et al., 2010). In contrast to the sequence autoencoder, the set autoencoder needs to be permutation invariant, both in the input and the output set. The first can be achieved directly by employing a recurrent encoder architecture using content-based attention similar to the one proposed by (Vinyals et al., 2015a) (see Section 3.1). Achieving permutation invariance in the output set is not straightforward. When training a set autoencoder, we need to provide the desired outputs $Y$ in some order. By definition, all possible orders should be equally good, as long as all elements of the input set and no surplus elements are present. In theory, the order in which the outputs are presented to the model (or, more specifically: to the loss function) should be irrelevant: by using a chain rule-based model, the RNN can, in principle, model any joint probability distribution, and we could simply enlarge the data set by including all permutations. Since the number of permutations grows exponentially in the number of elements, this is not a feasible way: The data set quickly becomes huge, and the model has to learn to create every possible permutation of output sets. Therefore, we need to tackle the problem of random permutations in the outputs differently, while maintaining a differentiable architecture (see Section 3.2).[2]

## 3.1 ENCODER: INPUT SET TO EMBEDDING

The encoder takes the input set and embeds it into the fixed-sized latent representation. This representation should be permutation invariant to the order of the inputs. We use an architecture with content-based attention mechanism similar to the one proposed in (Vinyals et al., 2015a) (see Figure 2):

$$m_i = f^{\text{inp}}(x_i) \tag{1}$$

$$c_t, h_t = \text{LSTM}(c_{t-1}, h_{t-1}, r_{t-1}) \tag{2}$$

$$e_{i,t} = f^{\text{dot}}(m_i, h_t) \tag{3}$$

$$a_{i,t} = \frac{\exp\left(e_{i,t}\right)}{\sum_j \exp\left(e_{j,t}\right)} \tag{4}$$

$$r_t = \sum_i a_{i,t} m_i \tag{5}$$

Figure 2: Encoder model

First, each element $x_i$ of the input set $X$ is mapped to a memory slot $m_i \in \mathbb{R}^l$ using a mapping function $f^{\text{inp}}$ (Eq. 1). We use a linear mapping as $f^{\text{inp}}$, the same for all $i$[3]. Then, an LSTM network (Hochreiter & Schmidhuber, 1997; Gers & Schmidhuber, 2000) with $l$ cells performs $n$ steps of calculation. In each step, it calculates its new cell state $c_t \in \mathbb{R}^l$ and hidden state $h_t \in \mathbb{R}^l$ using the previous cell- and hidden state $c_{t-1}$ and $h_{t-1}$, as well as the previous read vector $r_{t-1}$, all of which are initialized to zero in the first step. The new read vector $r_t$ is then calculated as weighted combination of all memory locations, using the attention mechanism (Eq. 5). For each memory location $i$, the attention mechanism calculates a scalar value $e_{i,t}$ which increases based on the similarity between the memory value $m_i$ and the hidden state $h_t$, which is interpreted as a query to the memory (Eq. 3). We set $f^{\text{dot}}$ to be a dot product. Then, the normalized weighting $a_i$ for all memory locations is calculated using a softmax (Eq. 4). After $n$ steps, the concatenation of $c_n, h_n$ and $r_n$ can be seen as a fixed-size embedding of $X$[4]. Note that permuting the elements of $X$ has no effect on the embedding, since the memory locations $m_i$ are weighted by content, and the sum in Eq. 5 is commutative.

---

[2]Note that, like the encoder, the decoder is implemented as a recurrent architecture. This is to accommodate for the fact that the set size is not specified – using a simple feed-forward architecture is not straight forward in this case.

[3](Vinyals et al., 2015a) used a "small neural network" for this task

[4]Preliminary experiments showed that the quality of the embedding (based on decoder performance) was relatively robust to the usage of different combinations of $c_n$, $h_n$, and $r_n$. We chose to include all three values, since it is not our goal to create the most compact embedding possible.

## 3.2 DECODER: EMBEDDING TO OUTPUT SET

Section 3.1 defined the forward propagation from the input set $X$ to the fixed-size embedding $[c_t, h_t, r_t]$. We now define the output of the set autoencoder that allows us to train the model using a loss function $\mathcal{L}$. Like in the original sequence-to-sequence model, the core of the decoder is an LSTM network (see Figure 3):

$$\hat{c}_t, \hat{h}_t = \text{LSTM}(\hat{c}_{t-1}, \hat{h}_{t-1}, \hat{r}_{t-1}) \qquad (6)$$

$$\hat{c}_0 = c_n \qquad (7)$$

$$\hat{h}_0 = h_n \qquad (8)$$

$$\hat{r}_0 = r_n, \quad \hat{r}_{t>0} = 0 \qquad (9)$$

$$o_t = f^{\text{out}}(\hat{h}_t) \qquad (10)$$

$$\omega_t = f^{\text{eos}}(\hat{h}_t) \qquad (11)$$

Figure 3: Decoder model

In each step, the decoder LSTM calculates its internal cell state $\hat{c}_t$ and its hidden state $\hat{h}_t$ (Eq. 6). The cell- and hidden state are initialized with the cell- and hidden state from the embedding, produced by the encoder (Eq. 7 and Eq. 8). In the first step, the decoder also gets an additional input $\hat{r}_0$, which is set to be the last read vector of the encoder (Eq. 9). In all following steps, $\hat{r}_t$ is a vector of all zeros. We calculate the decoder's output $o_t$ at step $t$ by using the linear function $f^{\text{out}}$ (Eq. 10). Each output element $o_t$ is of the same dimensionality as the input elements $x_i$. The underlying idea is that $f^{\text{out}}$ is the "reverse" operation to $f^{\text{inp}}$, such that encoder and decoder LSTM can operate on similar representations. Furthermore, in each step, the function $f^{\text{eos}}$ calculates $\omega_t$ (Eq. 11), which we interpret as the probability that $o_t$ is an element of the set. In other words, if $\omega_t = 0$, we can stop sampling.

In principle, we could use the LSTM's output sequence $O = (o_1, o_2, \ldots, o_m)$ directly as elements of the output set. However, this does not take into account the following issues: First, the number $m$ of outputs produced by the decoder should be equal to the size $n$ of the input set $X$. This can be achieved by learning to output the correct $\omega_t$ (see Eq. 12 below). Second, the order of the set elements should be irrelevant, i.e., the loss function should be permutation-invariant in the outputs. To address the latter point, we introduce an additional mapping layer $D = (d_0, d_1, \ldots, d_n)$ between the decoder output and the loss function. The mapping rearranges the first $n$ decoder outputs in the order of the inputs $X$. That is, it should make sure that the distance between $d_i$ and $x_i$ is small for all $i$. The mapping is defined as: $d_i = \sum_{j=1}^{n} o_j w_{ij}$ Here, $w_{ij}$ are the elements of a permutation matrix $W$ of size $n \times n$ with the properties

$$w_{ij} \in \{0, 1\} \ \forall i, j \in 1 \ldots n \qquad \sum_i w_{ij} = 1 \ \forall j \qquad \sum_j w_{ij} = 1 \ \forall i$$

In other words, each output $o_i$ is mapped to exactly one $d_j$, and vice versa. For now, we assume that $W$ has been parametrized such that the re-ordered elements in $D$ match the elements in input set $X$ well (see Section 3.3). Then, the set autoencoder loss function can be calculated as

$$\mathcal{L} = \sum_{i=1}^{n} L(x_i, d_i) + \sum_{i=1}^{m} L^{\text{eos}}(\omega_i, \omega_i^*) \qquad (12)$$

The function $L(x_i, d_i)$ is small if $x_i$ and $d_i$ are similar, and large if $x_i$ and $d_i$ are dissimilar. In other words, for each element in the input set, the distance to a matching output element (as mapped by $W$) will be decreased by minimizing $L$. For discrete elements, $L$ can be calculated as the cross entropy loss $L(x, d) = -\sum_i x_i \log d_i$. For elements that are vectors of real numbers, $L$ is a norm of these vectors, e.g., $L(x, d) = ||x - d||$. The function $L^{\text{eos}}$ calculates the cross-entropy loss between $\omega_i$ and $\omega_t^*$, where $\omega_i^*$ indicates if an $i^{\text{th}}$ element should be present in the output, i.e., $\omega_i^* = 1$ if $i <= n, 0$ else (recall that the decoder can produce $m$ outputs, and $m$ is not necessarily equal to $n$). Since the whole set autoencoder is differentiable, we train all weights except $W$ using gradient descent.

## 3.3 PARAMETRIZATION OF $W$: A STABLE MARRIAGE

Re-ordering the decoder outputs resembles a point cloud correspondence problem from the domain of computer vision (Jahne, 2000; Sonka et al., 2014). Methods like the iterative closest points algorithm

**Algorithm 1** Gale-Shapely algorithm for stable matching

Initialize all m ∈ M and w ∈ W to free
**while** ∃ free man m who still has a woman w to propose to **do**
    w = first woman on m's list to whom m has not yet proposed
    **if** w is free **then**
        (m, w) become engaged
    **else**
        **if** w prefers m to m*, to whom she is engaged **then**
            m* becomes free
            (m, w) become engaged

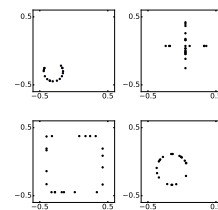

Figure 4: Examples of sets in *shapes* data set

(Besl & McKay, 1992) find closest points between two sets, and find a transformation that aligns the second set to the first. Since we are only interested in the correspondence step, we notice its similarity to the stable marriage problem (Gusfield & Irving, 1989): We want to find matching pairs $P^i = \{\text{man}_i, \text{woman}_i\}$ of two sets of men and women, such that there are no two pairs $P^i$, $P^j$ where element $\text{man}_i$ prefers $\text{woman}_j$ over $\text{woman}_i$, and, at the same time, $\text{woman}_j$ prefers $\text{man}_i$ over $\text{man}_j$.[5] To solve this problem greedily, we can use the Gale-Shapely algorithm (Gale & Shapley, 1962), which has a run time complexity of $\mathcal{O}(n^2)$ (see Algorithm 1)[6]. Since its solution is permutation invariant in the set that proposes first (Gusfield & Irving, 1989)(p. 10), we consider the input elements $x_i$ to be the men, and let them propose first. After the stable marriage step, $w_{ij} = 1$ if $x_i$ is "engaged" to $o_j$.

## 4 EXPERIMENTS

### 4.1 UNSUPERVISED LEARNING OF EMBEDDING

We use a number of synthetic data sets of point clouds for the unsupervised experiments. Each data set consists of sets of up to $k$ items with $d$ dimensions. In the *random* data sets, the $k$ values of each element are randomly drawn from a uniform distribution between -0.5 and 0.5. In the following experiments, we set $k = 16$ and $d \in \{1, 2, 3\}$. In other words, we consider sets of up to 16 elements that are randomly distributed along a zero-centered 1d-line, 2d-plane, or 3-d cube with side length 1. We choose random distributions to evaluate the architecture's capability of reconstructing elements from sets, rather than learning common structure within those sets. In the *shapes* data set, we create point clouds with up to $k = 32$ elements of $d = 2$ dimensions. In each set, the points form either a circle, a square, or a cross (see Figure 4). The shapes can occur in different positions and vary in size. To convey enough information, each set consists of at least 10 points. Each data set contains 500k examples in the training set, 100k examples in the validation set, and another 500k examples in the test set.

For each of the data sets we train a set autoencoder to minimize the reconstruction error of the sets, i.e., to minimize Eq. 12 (please refer to the appendix for details of the training procedure, including all hyperparameters). Figure 5 shows the mean euclidean distance of the reconstructed elements for the three *random* data sets (left-hand side) and the *shapes* data set (right-hand side), for different set sizes. For the random data sets, the mean error increases with the number of dimensions $d$ of the elements, and with the number of elements within a set. This is to be expected, since all values are completely uncorrelated. For the shapes data set, the average error is lower than the errors for the 2d-random data set with more than two elements. Furthermore, the error decreases with the number of elements in the set. We hypothesize that with a growing number of points, the points become more evenly distributed along the outlines of the shapes, and it is therefore easier for the model to reconstruct them (visual inspection of the reconstructions suggests that the model tends to distribute points more uniformly on the shapes' outlines).

We now take a closer look at the embeddings of the set autoencoder (i.e., the vector $[c_t, h_t, r_t]$)

---

[5]Note that this is a relaxation of the correspondence problem, since the concept of *preference* in the stable marriage problem is ordinal rather than cardinal, i.e., we do *not* consider the exact distances between elements, but ranks.

[6]This complexity could restrict the applicability of the proposed algorithm to smaller sets. However, there is a range of problems where small set sizes are relevant, e.g. when an agent interacts with an environment where one or multiple instances of an object can be present (as opposed to point cloud representations of objects).

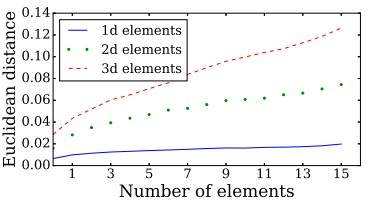 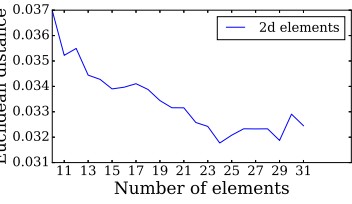

Figure 5: Mean reconstruction error of reconstructed elements in sets of different size.

(a) Random data sets with $d \in \{1, 2, 3\}$      (b) Shapes data set ($d = 2$)

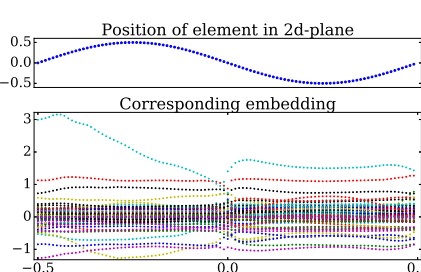

Figure 6: Smoothness of embeddings for sets with a single 2d-element. Each vertical slice through both graphs represents data for a single set. The top shows different positions of the element in in a $1 \times 1$ plane, the bottom the corresponding embeddings. E.g., the first point from the left in the top diagram represents a set of size 1, with a single element at the coordinates $[0, -0.5]$. The points directly below visualize the values of the embedding corresponding to this set. Best viewed in color.

for random sets. Some of the embedding variables have a very strong correlation with the set size (Pearson correlation coefficients of $> 0.97$ and $<$-0.985, respectively). In other words, the size of the set is encoded almost explicitly in the embedding.

The embeddings seem to be reasonably smooth (see Figure 6). We take a set with a single 2d-element and calculate its embeddings (leftmost points in Figure 6). We then move this element smoothly in the 2d-plane and observe the resulting embeddings. Most of the time, the embeddings change smoothly as well. The discontinuities occur when the element crosses the center region of the 2d plane. Apart from this center region, the embedding preserves a notion of *distance* of two item sets. This becomes apparent when looking at the correlations between the euclidean distances of two sets $X^1$ and $X^2$ and their corresponding embeddings $enc(X^1)$ and $enc(X^2)$.[7] The correlation coefficients for random sets of size one to four are 0.81, 0.71, 0.67, and 0.64. In other words, similar sets tend to yield similar embeddings.

Vinyals et al. show that order matters for sequence-to-sequence and set-to-set models. This is the case both for the order of input sequences – specific orders improve the performance of the model's task – as well as for the order of the output sets, i.e., the order in which the elements of the set are processed to calculate the loss function. Recall that the proposed set autoencoder is invariant to the order of the elements both in the input set (using the attention mechanism) and the target set (by reordering the outputs before calculating the loss). Nevertheless, we observe that the decoder learns to output elements in a particular order: We now consider sets with exactly 8 random 2-dimensional elements. We use a pretrained set autoencoder from above, encode over 6,000 sets, and subsequently reconstruct the sets using the decoder. Figure 7 shows heat maps of the 2d-coordinates of the $i$'th reconstructed element. The first reconstructed element tends to be in the center right area. Accordingly, the second element tends to be in the lower-right region, the third element in the center-bottom region, and so on. The decoder therefore has learned to output the elements within a set in a particular order. Note that most of the distributions put some mass in every area, since the decoder must be able to reproduce sets where the items are not distributed equally in all areas (e.g., in a set where all elements are in the top right corner, the first element must be in the top right corner as well). Figure 8 shows the effect of the set size $n$ on the distribution of the first element. If there is only one element ($n = 1$), it can be anywhere in the 2d plane. The more elements there are, the more likely it is that at least one of them will be in the center right region, so the distribution of the first element gets more peaked. We speculate that the roughly circular arrangement of the element distributions (which occurs for other set sizes as well) could be an implicit result of using the cosine similarity $f^{dot}$ in the attention mechanism of the encoder. Also, this behavior is likely to be the reason for the discontinuity in Figure 6 around $[0, 0]$.

---

[7]We align the elements of $X^1$ and $X^2$ using the Gale-Shapely algorithm before calculating distances.

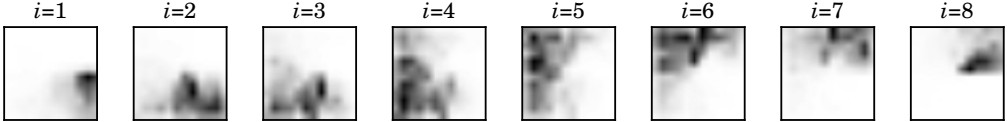

Figure 7: Heat maps of the location of the $i$th element in sets of size 8 with 2d-elements (decoder output). Darker shadings indicate more points at these coordinates.

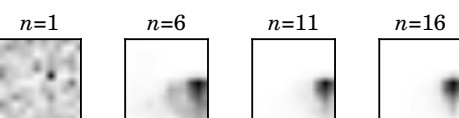

Figure 8: Heat maps of the location of the first element in sets of various sizes $n$ with 2d-elements (decoder output). Darker shadings indicate more points at these coordinates.

## 4.2 SUPERVISED LEARNING: CLASSIFICATION AND REGRESSION TASKS

We derive a number of classification and regression tasks based on the data sets in Section 4.1. On the *random* data sets, we define a number of binary classification tasks. The 1d-, 2d- or 3d- area is partitioned into two, four, or eight areas of equal sizes. Then, two classes of sets are defined: A set is of class 1, if all of its elements are within $i$ of the $j$ defined areas. All other sets are of class two. For example, if $d = 2$, $i = 2$, and $j = 4$, a set is of class 1 if all its elements are in the top left or bottom left area, or any other combination of two areas.[8] Furthermore, we define two regression tasks on the random data sets: The target for the first one is the *maximum distance* between any two elements in the set, the second one is the volume of the $d$-dimensional *bounding box* of all elements. On the *shapes* data set, the three-class classification problem is to infer the prototypical shape represented by the elements in the set.

In the following, we use a set autoencoder as defined above, add a standard two-layer neural network $f^{\text{supervised}}(\text{enc}(X))$ on top of the embedding, and use an appropriate loss function for the task (for implementation details see supplementary material). We compare the results of the set autoencoder (referred to as set-AE) to those of two vanilla sequence autoencoders, which ignore the fact that the elements form a set. The first autoencoder (seq-AE) gets the same input as the set model, the input for the second autoencoder has been ordered along the first element dimensions (seq-AE (ordered)). Furthermore, we consider three training fashions: For *direct* training, we train the respective model in a purely supervised fashion on the available training data (in other words: the models are *not* trained as autoencoders). In the *pretrained* fashion (*pre*), we initialize the encoder weights using unsupervised pretraining on the respective (unlabeled) 500k training set, and subsequently train the parameters of $f^{\text{supervised}}$, holding the encoder weights fixed. In the *fine tuning* setting (*fine*), we continue training from the *pretrained* model, and fine-tune all weights.

Tables 1(a) and (b) show the accuracy on the test set for the $i$-of-$j$-areas classification tasks, for 1,000 and 10,000 training examples. The plain sequence autoencoder is only competitive for the most simple task (first row). For the simple and moderately complex tasks (first three rows), the ordered sequence autoencoder and the set autoencoder reach high accuracies close to 100%, both

| | | Seq-AE | | | Seq-AE (ordered) | | | Set-AE | | |
|---|---|---|---|---|---|---|---|---|---|---|
| areas | d | direct | pre | fine | direct | pre | fine | direct | pre | fine |
| 1 of 2 | 1 | 0.978 | 0.909 | 0.996 | 0.984 | 0.994 | 0.998 | 0.995 | 0.998 | **0.999** |
| | 2 | 0.971 | 0.915 | 0.952 | 0.979 | 0.971 | **0.996** | 0.990 | 0.981 | 0.983 |
| 1 of 4 | 1 | 0.948 | 0.752 | 0.968 | 0.949 | 0.973 | **0.986** | 0.962 | 0.974 | 0.980 |
| | 2 | 0.963 | 0.908 | 0.953 | 0.962 | 0.931 | 0.963 | 0.986 | 0.985 | **0.988** |
| 1 of 8 | 3 | 0.955 | 0.887 | 0.939 | 0.964 | 0.935 | 0.950 | 0.973 | 0.973 | **0.978** |
| 2 of 4 | 1 | 0.754 | 0.554 | 0.661 | 0.847 | 0.884 | 0.940 | 0.883 | 0.944 | **0.954** |
| | 2 | 0.777 | 0.660 | 0.704 | 0.872 | 0.766 | 0.800 | 0.935 | 0.941 | **0.949** |
| 2 of 8 | 3 | 0.772 | 0.541 | 0.638 | 0.853 | 0.738 | 0.827 | 0.869 | **0.908** | **0.908** |
| 4 of 8 | 3 | 0.656 | 0.537 | 0.527 | 0.672 | 0.543 | 0.597 | 0.689 | 0.752 | **0.764** |

(a) 1,000 training examples

| | | Seq-AE | | | Seq-AE (ordered) | | | Set-AE | | |
|---|---|---|---|---|---|---|---|---|---|---|
| areas | d | direct | pre | fine | direct | pre | fine | direct | pre | fine |
| 1 of 2 | 1 | 0.990 | 0.912 | 0.998 | 0.998 | 0.997 | **1.000** | 0.999 | 0.998 | 0.999 |
| | 2 | 0.983 | 0.909 | 0.978 | 0.995 | 0.980 | **0.998** | **0.998** | 0.985 | 0.991 |
| 1 of 4 | 1 | 0.987 | 0.783 | 0.990 | 0.993 | 0.985 | **0.997** | 0.990 | 0.978 | 0.995 |
| | 2 | 0.984 | 0.918 | 0.985 | 0.991 | 0.956 | 0.990 | **0.997** | 0.991 | 0.995 |
| 1 of 8 | 3 | 0.983 | 0.857 | 0.982 | 0.986 | 0.950 | 0.984 | **0.995** | 0.987 | 0.993 |
| 2 of 4 | 1 | 0.933 | 0.594 | 0.864 | 0.970 | 0.925 | 0.985 | **0.992** | 0.969 | 0.985 |
| | 2 | 0.919 | 0.545 | 0.796 | 0.950 | 0.815 | 0.946 | **0.986** | 0.969 | 0.980 |
| 2 of 8 | 3 | 0.924 | 0.535 | 0.820 | 0.938 | 0.800 | 0.918 | 0.962 | 0.960 | **0.971** |
| 4 of 8 | 3 | 0.720 | 0.535 | 0.631 | 0.758 | 0.562 | 0.716 | 0.810 | 0.860 | **0.888** |

(b) 10,000 training examples, same rows as (a)

Table 1: Accuracy for area classification tasks, higher is better. Averaged over 10 runs.

---

[8]We create new labeled data sets with the same number of examples per class

**(a) 1,000 training examples**

| task | d | Seq-AE direct | pre | fine | Seq-AE (ordered) direct | pre | fine | Set-AE direct | pre | fine |
|---|---|---|---|---|---|---|---|---|---|---|
| boun- | 1 | 0.079 | 0.140 | 0.084 | 0.043 | 0.058 | **0.021** | 0.054 | 0.059 | 0.038 |
| ding | 2 | 0.085 | 0.133 | 0.124 | 0.075 | 0.110 | 0.088 | **0.049** | 0.073 | 0.062 |
| box | 3 | 0.083 | 0.119 | 0.120 | 0.082 | 0.110 | 0.095 | **0.074** | 0.090 | 0.087 |
| max | 1 | 0.080 | 0.140 | 0.083 | 0.043 | 0.057 | **0.018** | 0.042 | 0.059 | 0.039 |
| dis- | 2 | 0.099 | 0.142 | 0.121 | 0.089 | 0.123 | 0.104 | **0.059** | 0.076 | 0.061 |
| tance | 3 | 0.114 | 0.141 | 0.140 | 0.111 | 0.132 | 0.124 | **0.083** | 0.107 | 0.100 |

**(b) 10,000 training examples, same rows as (a)**

| Seq-AE direct | pre | fine | Seq-AE (ordered) direct | pre | fine | Set-AE direct | pre | fine |
|---|---|---|---|---|---|---|---|---|
| 0.024 | 0.140 | 0.020 | 0.015 | 0.052 | **0.007** | 0.017 | 0.054 | 0.016 |
| 0.032 | 0.133 | 0.048 | 0.023 | 0.106 | 0.020 | 0.019 | 0.067 | **0.017** |
| 0.067 | 0.119 | 0.095 | 0.046 | 0.109 | 0.056 | **0.022** | 0.088 | 0.047 |
| 0.023 | 0.140 | 0.023 | 0.011 | 0.052 | **0.007** | 0.016 | 0.055 | 0.015 |
| 0.042 | 0.140 | 0.071 | 0.031 | 0.117 | 0.034 | **0.017** | 0.069 | 0.022 |
| 0.065 | 0.140 | 0.096 | 0.055 | 0.131 | 0.072 | **0.022** | 0.103 | 0.049 |

Table 2: RMSE on Regression tasks, lower is better . Averaged over 10 runs.

**(a) 1,000 training examples**

| Seq-AE direct | pre | fine | Seq-AE (ordered) direct | pre | fine | Set-AE direct | pre | fine |
|---|---|---|---|---|---|---|---|---|
| 0.531 | 0.402 | 0.428 | 0.667 | 0.779 | **0.809** | 0.602 | 0.634 | 0.641 |

**(b) 10,000 training examples, same rows as (a)**

| Seq-AE direct | pre | fine | Seq-AE (ordered) direct | pre | fine | Set-AE direct | pre | fine |
|---|---|---|---|---|---|---|---|---|
| 0.580 | 0.384 | 0.548 | 0.849 | 0.838 | **0.911** | 0.697 | 0.699 | 0.732 |

Table 3: Accuracy for object shape classification task, higher is better. Averaged over 10 runs.

for the small and the large training set. When the task gets more difficult (higher $i$, $j$, or $d$), the set autoencoder clearly outperforms both other models. For the small training set, the *pre* and *fine* training modes of the set autoencoder usually lead to better results than *direct* training. In other words, the unsupervised pretraining of the encoder weights leads to a representation which can be used to master the classification tasks with a low number of labeled examples. For the larger training set, unsupervised pretraining is still very useful for the more complicated classification tasks. On the other hand, unsupervised pretraining only helps in a few rare cases if the elements are treated as a sequence – the representation learned by the sequence autoencoders does not seem to be useful for the particular classification tasks.[9]

The results or the regression task are shown in Tables 2 (a) and (b). Again, the ordered sequence autoencoder shows good results for small $d$ (recall that the first element dimension is the one that has been ordered), but fails to compete with the set-aware model in the higher dimensions. However, unsupervised pretraining helps the set model in the regression task only for small $d$.

Tables 3 (a) and (b) show the results for the *shapes* classification task. Here, the ordered sequence autoencoder with fine tuning clearly dominates both other models. The set model is unable to capitalize on the proper handling of permutation invariance.

In sum, the results show that unsupervised pretraining of the set autoencoder creates representations that can be useful for subsequent supervised tasks. This is primarily the case, if the supervised task requires knowledge of the individual locations of the elements, as in the $i$-of-$j$-areas classification task. If the *precise* locations of a subset of the elements are required (as in the *bounding box* or *maximum distance* regression tasks), direct training yields better results. We hypothesize that failure of the set-aware model on the *shapes* classification is due to the linear mapping functions $f^{\text{inp}}$ and $f^{\text{out}}$: They might be too simple to capture the strong, but non-linear structures in the data.

## 5 CONCLUSION

We presented the set autoencoder, a model that can be trained to reconstruct sets of elements using a fixed-size latent representation. The model achieves permutation invariance in the inputs by using a content-based attention mechanism, and permutation invariance in the outputs, by reordering the outputs using a stable marriage algorithm during training. The fixed-size representation possesses a number of interesting attributes, such as distance preservation. We show that, despite the output permutation invariance, the model learns to output elements in a particular order. A series of experiments show that the set autoencoder learns representations that can be useful for tasks that require information about each set element, especially if the tasks are more difficult, and few labeled training examples are present. There are a number of directions for future research. The most obvious is to use non-linear functions for $f^{\text{inp}}$ and $f^{\text{out}}$ to enable the set autoencoder to capture non-linear structures in the input set, and test the performance on point clouds of 3d data sets such as ShapeNet (Chang et al., 2015). Also, changes to the structure of the encoder/decoder (e.g., which variables are

---

[9]This is despite the fact that the reconstruction error after the unsupervised training is *much lower* for the sequence autoencoders than for the set autoencoder (not in the results tables).

interpreted as query or embedding) and alternative methods for aligning the decoder outputs to the inputs can be investigated. Furthermore, more research is necessary to get a better understanding for which tasks the permutation invariance property is helpful, and unsupervised pretraining can be advantageous.

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

## APPENDIX

## MODEL DETAILS AND TRAINING PROCEDURE

### ARCHITECTURE AND SIZING

We use Tensorflow v0.12 (Abadi et al., 2016) to implement all models. For the implementation and experiments, we made the following design choices:

**Model Architecture**

- Both the encoder and the decoder LSTMs are have peephole connections (Gers & Schmidhuber, 2000). We use the LSTM implementation of Tensorflow [10]

- The input mapping $f^{\text{inp}}$ and output mapping $f^{\text{out}}$ functions are simple linear functions. Note that we can *not* re-use $f^{\text{inp}}$ on the decoder side to transform the supervised labels in a "backwards" fashion: In this case, learning could parametrize $f^{\text{inp}}$ such that all set elements are mapped to a the same value, and the decoder learns to output this element only.

- For the supervised experiments (classification and regression), we add a simple two-layer neural network $f^{\text{supervised}}$ on top of the embedding. The hidden layer of this network has the same number of units as the embedding, and uses ReLu activations (Nair & Hinton, 2010). For the two-class problems ($i$ of $j$ areas), we use a single output neuron and a cross-entropy loss, for the multi-class problems (*object shapes*) we use three output neurons and a cross-entropy loss. For the regression problems (*bounding box* and *maximum distance*), we optimize the mean squared error.

- All parameters are initialized using Xavier initialization (Glorot & Bengio, 2010)

- Batch handling: For efficiency, we use minibatches. Therefore, within a batch, there can be different set sizes $n^i$ for each example $i$ in the set. For simplicity, the encoder keeps processing all sets in a batch, i.e., it always performs $n = k$ steps, where $k$ is the maximum set size. Preliminary experiments showed only minor variations in the performance when processing is stopped after $n^i$ steps, where $n^i$ corresponds to the actual size of set $i$ in the minibatch.

**Dimensionality of layers**

- The number $l$ of LSTM cells is automatically determined by the dimensionality $d$ and maximum set size $k$ of the input. We set $l = k * d$, therefore $c_t, h_t, \hat{c}_t, \hat{h}_t \in \mathcal{R}^l$. As a consequence, the embedding for all models (set- and sequence autoencoder) could, in principle, comprise the complete information of the set (recall that the goal was not to find the most compact or efficient embedding)

- For simplicity, the dimensionality of each the memory cell $m_i$ and the read vector $r_i$ is equal to the number of LSTM cells, i.e., $m_i, r_i \in \mathcal{R}^l$ (in principle, the memory could have any other dimensionality, but this would require an additional mapping step, since the query needs to be of the same dimensionality as the memory).

### TRAINING

We use Adam (Kingma & Ba, 2014) to optimize all parameters. We keep Adam's hyperparameters (except for the learning rate) at Tensorflow 's default values ($\beta_1 = 0.9$, $\beta_2 = 0.999$, $\epsilon = 1e-08$). We use minibatch training with a batch size of 100. We keep track of the optimization objective during

---

[10]https://github.com/tensorflow/tensorflow/blob/r0.12/tensorflow/python/ops/rnn_cell.py#L363

| Training mode | initial $\alpha$ | stalled epochs before decreasing $\alpha$ | stalled epochs before stopping | Weight decay $\delta$ |
|---|---|---|---|---|
| Unsup. pretraining | 0.002 | 20 | 40 | 0.0 for all params |
| Supervised, *direct* | 0.0002 | 200[11] | 400 | 0.001 for all params |
| Supervised, *pre* | 0.002 | 200 | 400 | 0.001 for $f^{\text{supervised}}$ |
| Supervised, *fine* | 0.0002 | 200 | 400 | 0.001 for $f^{\text{supervised}}$ |

Table 4: Training hyperparameters.

training and reduce the learning rate by 33% / stop training once there has been no improvement for a defined number of epochs, depending on the training mode (see Table 4). For the classification tasks, we couple the learning rate decrease/early stopping mechanism to the missclassification error $(1 - \text{accuracy})$ rather than the loss function.

*Remarks*

- The values of *stalled-epochs-before-X* are much higher for the supervised learning scenarios, since the training sets are much smaller (e.g., when using 1,000 examples and a batch size of 100, a single epoch only results in 10 gradient update steps).

- It is possible that the results for supervised training with fine tuning improve if the encoder weights are regularized as well (the weights are prone to overfitting, since we use a low number of training examples).

