# OpenReview forum: "The Set Autoencoder: Unsupervised Representation Learning for Sets"
_ICLR.cc/2018/Conference — Reject_

### Official Review · AnonReviewer1 · 2017-11-24

**Rating:** 4
**Confidence:** 5

**Review:**

This paper mostly extends Vinyals et al, 2015 paper ("Order Matters") on how to represent sets as input and/or output of a deep architecture.

As far as I understood, the set encoder is the same as the one in "Order Matters". If not, it would be useful to underline the differences.

The decoder, on the other hand, is different and relies on a loss that is based on an heuristic to find the current best order (based on an ordering, or mapping W, found using the Gale-Shapely algorithm). Does this mean that Algorithm 1 needs to be run for every training (and test) example? if so, it is important to note what is the effective complexity of running it?

The experimental section is interesting, but in the end a bit disappointing: although a new artificial dataset is proposed to evaluate sets, it is unclear how different are the findings from those in the "Order Matters" paper:
- the first set of results (in Section 4.1) confirms that the set encoder is important (which was also in the other paper I believe)
- the second set of results (Section 4.2) shows that in some cases, an auto-encoder is also useful: this is mostly the case when the supervised data is small compared to the availability of a much larger unsupervised data (of sets). This is interesting (and novel compared to the "Order Matters" paper) but corresponds to known findings from most previous work on semi-supervised learning: pre-training is only useful when only a very small supervised data exists, and quickly becomes irrelevant. This is not specific to sets.

Finally, It would have been very interesting to see experiments on real data concerned with sets.

------------------
I have read the respond to the reviewers but haven't seen any reason to
change my score. In particular, the authors have not answered my questions
about differences with the prior art, and have not provided results on
real data.

---

> ### Author Response · Authors · 2018-01-10
> **prior art**
>
> Thank you for your comment.
>
> Regarding your question about prior art: Yes, the encoder is conceptually identical to the one proposed in the "Oder Matters" paper. We wrote "similar to" in the initial version since the some of the architectural details are not completely disclosed in the "Order Matters" paper (e.g. the "small neural network" for f^inp, which probably uses non-linearities, whereas our f^inp is linear). But structurally, the encoder is identical.

---

### Official Review · AnonReviewer2 · 2017-11-27
**Good preliminary results**

**Rating:** 5
**Confidence:** 4

**Review:**

Summary
This paper proposes an autoencoder for sets. An input set is encoded into a
fixed-length representation using an attention mechanism (previously proposed by
[1]). The decoder generates the output sequentially and the generated sequence
is matched to the best-matching ordering of the target output set.
Experiments are done on synthetic datasets to demonstrate properties of the
learned representation.

Pros
- Experiments show that the autoencoder helps improve classification accuracy
  for small training set sizes on the shape classification task.
- The analysis of how the decoder generates data is insightful.

Cons
- The experiments are on toy datasets only. Given the availability of point
  cloud data sets, for example, KITTI which has a widely used benchmark for
point cloud based object detection, it would make the paper stronger if this
model was benchmarked against published baselines.

- The autoencoder does not seem to help much on the regression tasks where even
  for the smaller training set size setting, directly using the encoder to solve
the task often works best. Even finetuning is unable to recover from the
pretrained weights. Therefore, it seems that the decoder (which is the novel
aspect of this work) is perhaps not working well, or is not well suited to the
regression tasks being considered.

- The classification task, for which the learned representations work well
  empirically, seems to be geared towards representing object shape. It doesn't
really require remembering each point. On the other hand, the regression tasks
that could require remembering the points don't seem to be benefit much from the
autoencoder pretraining. This suggests that while the model is able to represent
overall shape, it has a hard time remembering individual elements of the set.
This seems like a drawback, since a general "set auto-encoder" should be able
to perform a wide variety of tasks on the input set which could require remembering
the set's elements.

Quality
This paper describes the proposed model quite well and provides encouraging
preliminary results.

Clarity
The paper is easy to understand.

Originality
The novelty in the model is using a matching algorithm to find the best ordering
of the target output set to match with the sequentially generated decoder
output. However, the paper makes a choice of one ranking based matching scheme
and does not compare to other alternatives.

Significance
This paper proposes a way of learning representations of sets which will be of
broad interest across the machine learning community. These models are likely to
become more relevant with increasing prevelance of point cloud data.

References
[1] Oriol Vinyals, Samy Bengio, and Manjunath Kudlur. Order matters: Sequence to
sequence for sets. arXiv preprint arXiv:1511.06391.

---

### Official Review · AnonReviewer3 · 2017-11-28
**An interesting framework but more real-world experiments needed**

**Rating:** 5
**Confidence:** 4

**Review:**

Summary:

This paper proposes an encoder-decoder framework for learning latent representations of sets of elements. The model utilizes the neural attention mechanism for set inputs proposed in (Vinyals et al., ICLR 2016) to encode a set into a fixed-length latent representation, and then employs an LSTM decoder to reconstruct the original set of elements, in which a stable matching algorithm is used to match decoder outputs to input elements. Experimental results on synthetic datasets show that the model learns meaningful representations and effectively handles permutation invariance.

Major Concerns:

1. Although the employed Gale-Shapely algorithm facilitates permutation-invariant set reconstruction, it has O(n^2) computational complexity during each back-propagation iteration, which might prevent it from scaling to sets of fairly big sizes.

2. The experiments are only evaluated on synthetic datasets, and applications of the set autoencoder to real-world applications or scientific problems will make this work more interesting and significant.

3. The main contribution of this work is the adoption of the stable matching algorithm in the decoder. A strong set autoencoder baseline will be, the encoder employs the neural attention mechanism proposed in (Vinyals et al., ICLR 2016), but the decoder just uses a standard LSTM as in a seq2seq framework. Comparisons to this baseline will reveal the contribution of the stable matching procedure in the whole  framework of  the set autoencoder for learning representations.

Minor issues:

On page 5, above Section 4, d_j -> o_j ?

the footnote on page 5: we not consider -> we do not consider?

on page 6 and 7,   6.000, 1.000 and 10.000 training examples ->  6000, 1000 and 10,000 training examples

---

### Author Response · Authors · 2017-12-19
**Authors' comments to reviews**

We thank the reviewers for the insightful and encouraging remarks. We comment on a number of these remarks below, and have updated some of the corresponding points in the paper.

== Major concerns of one or multiple reviewers ==

* O(n^2) complexity of Gale-Shapely.
It is true that this complexity could, in practice, restrict the applicability of the proposed algorithm to smaller sets. However, there is a range of problems where small set sizes are relevant, e.g. when an agent interacts with an environment where one or multiple instances of an object can be present (as opposed to point cloud representations of objects)

We have included the above remark in the paper.

* Synthetic data set vs. real-world data set
We completely agree that the paper will be much stronger once we include results on a real-world data set. However, in the limited time available, we were not able to do so just yet.

* Proposal by AnonReviewer3: use the same encoder, but a plain LSTM decoder as benchmark (to show whether the Gale-Shapely-augmented decoder works better).
(i.e., use the first $n$ outputs $o_i,i\in{1,\dots,n}$ directly)

This is an interesting idea, that we will have to try out. However, the current assumption is that its behavior will probably be worse: Unlike the Seq-AE, it will not be able to store ordering information in the permutation-invariant embedding, but penalize misaligned points in the output heavily.

* Remarks about applicability to different problem types

We agree with the reviewers' comments about the applicability of the model (and its limitations). The purpose of using a range of problems with different properties was precisely to test this. We currently think that future work could either try to make the model more generally applicable to multiple of these problem classes, or specialize it for a specific type of problem (however, we think that it would be beyond the scope of this paper, especially when taking the page limit into account).

== Minor issues raised by one or multiple reviewers ==
* We fixed multiple smaller issues (typos/formatting) in the latest version.

---

### Decision · Program_Chairs · 2018-01-29
**ICLR 2018 Conference Acceptance Decision**

**Decision:**

Reject

**Comment:**

The paper proposes an autoencoder for sets, an interesting and timely problem.  The encoder here is based on prior related work (Vinyals et al. 2016) while the decoder uses a loss based on finding a matching between the input and output set elements.  Experiments on multiple data sets are given, but none are realistic.   The reviewers have also pointed out a number of experimental comparisons that would improve the contribution of the paper, such as considering multiple matching algorithms and more baselines.  In the end the idea is reasonable and results are encouraging, but too preliminary at this point.